# Record linkage studies of drug-related deaths among former adult prisoners who have been released to the community: a scoping review protocol

Janine A Cooper  ,[1] Ifeoma Onyeka,[1] Dermot O'Reilly,[1] Richard Kirk,[2] Michael Donnelly[1]

¹Administrative Data Research Centre Northern Ireland (ADRC NI), Centre for Public Health, Queen's University Belfast, Royal Hospitals Site, Belfast, UK
²South Eastern Health and Social Care Trust, Ulster Hospital, Dundonald, UK

**Correspondence to**
Dr Janine A Cooper;
j.cooper@qub.ac.uk

## ABSTRACT

**Introduction** Prisoners often have a complex mix of health and care needs including needs relating to mental and physical ill-health conditions. In addition, substance use disorders are common in prisoners. The transition between prison and community poses risks in terms of a disruption to treatment and care and a deterioration in health. Studies have demonstrated an increased risk of drug-related deaths among former prisoners during the first weeks following release from custody. The aim of this scoping review is to provide an up-to-date account of evidence from record linkage studies about drug-related deaths among former adult prisoners.

**Methods and analysis** This is a protocol for a scoping review. The framework for conducting scoping reviews by Arksey and O'Malley, in addition to the methodology and guidance developed by the Joanna Briggs Institute (JBI) and the JBI Collaboration will be followed. MEDLINE, EMBASE, PsychINFO and Web of Science will be searched for studies from 2011 to 2021 using search terms relating to 'mortality', 'drugs' and 'ex-prisoner'. A preliminary search in MEDLINE identified at least 473 potentially eligible papers. Two authors will independently screen all titles and abstracts for eligibility using the inclusion and exclusion criteria. There will be no geographical restrictions but non-English language papers will be excluded. Data will be extracted using a piloted customised charting form and the findings will be mapped (guided by the population, concept and context inclusion criteria) and grouped by main conceptual categories (including methodology, key findings and gaps in the research).

**Discussion, ethics and dissemination** This scoping review will identify, map and summarise publicly available sources of research and therefore does not require ethical approval. The results will inform our empirical investigation of ex-prisoner mortality as well as being shared with key stakeholders, disseminated in a peer-reviewed journal and presented at relevant conferences.

---

### Strengths and limitations of this study

► The proposed methodology for conducting our scoping review, including the use of the framework by Arksey and O'Malley, and methodology and guidance by the Joanna Briggs Institute is outlined by this protocol.
► The Preferred Reporting Items for Systematic reviews and Meta-Analyses extension for Scoping Reviews checklist and guidance will be used to report the proposed review.
► MEDLINE, EMBASE, PsychINFO and Web of Science will be searched for studies from 2011 to 2021 using search terms relating to 'mortality', 'drugs' and 'ex-prisoner'.
► There will be no geographical restrictions but non-English language papers will be excluded.
► The proposed scoping review will map research evidence in this area, and may help identify and profile former prisoners who are most at risk, and inform potential interventions and future research and policy.

## INTRODUCTION

The world prison population is estimated to be more than 11 million.[1] Prison population rates vary by regions; the world prison population rate is 145 per 100 000, however, the highest rate of 655 per 100 000 is found in the USA with over 2.1 million prisoners incarcerated.[1] Prisoners often have a complex mix of health and care needs including needs relating to mental and physical ill-health conditions, homelessness, unemployment and substance addictions.[2] Compared with the general population, prisoners have more physical and mental health problems and prison confinement serves as an opportunistic setting for the healthcare of this population.[3 4] The provision of care to this population is expected to be equivalent to the care that is delivered to the wider community, and according to the United Nations (1990) *Basic Principles for the Treatment of Prisoners*, prisoners are entitled to access health services available in their resident country without discrimination.[5] However, the transition between prison and community including

the transfer of care poses risks in terms of a disruption to treatment and care and a deterioration in health.[4]

Mortality risk among former prisoners has been extensively studied using epidemiological methodology. Previous systematic reviews and meta-analyses among ex-prisoners have found an increased risk of death following release from prison, however, reviews have been limited by high levels of heterogeneity. A systematic review, published in 2012, explored the use of record linkage methods to examine deaths in ex-prisoners between 1998 and 2011, finding 29 publications (based on 25 studies).[6] The methodological review found limitations in both study quality and high levels of heterogeneity in the study design, analysis and findings of eligible publications.[6] Furthermore, most included studies were conducted in the USA, UK or Australia, therefore limiting the generalisability of the findings.[6] The review found an elevated risk of all-cause mortality, unnatural death, suicide and drug-related death among ex-prisoners for more than 1 year after release.[6] A separate systematic review and meta-analysis of publications between 1980 and 2011 explored the literature on studies of mortality in released prisoners using linkage of prisoner and mortality databases.[7] There is a high level of overlap in publications included in these systematic reviews.[6 7] The review, of 18 cohorts, totalling more than 400 000 released prisoners and 26 163 deaths, was again limited by high heterogeneity of death rates as a result of differences in the study designs of primary publications.[7] The review found an increased risk of death from all causes, and in particular from suicide, homicide and drugs, with approximately one-fifth of deaths caused by drugs.[7]

Substance use disorders are common in prisoners with approximately a third of male and a half of female prisoners entering prison with a drug use disorder.[8] Studies have demonstrated an increased risk of drug-related deaths among ex-prisoners following release from custody.[9–11] A meta-analysis of deaths among former prisoners (studies published between 1966 and 2009), including six studies (n=1033 deaths), reported that 59% of deaths within the first 3 months of release were due to drug-related causes; this increased to 76% in the first 2 weeks after release.[12] The review focused on exploring mortality during the first 12 weeks following release from prison only and data were extracted from eligible publications for this period.[12] The meta-analysis reported a threefold to eightfold increase in risk of drug-related death in the first 2 weeks after release from prison compared with the following 10 weeks (although the risk remained elevated for the first 4 weeks).[12] However, in the meta-analysis, heterogeneity was present between countries (USA, UK or Australia).[12] There is a high level of overlap in publications included in the meta-analysis and in the reviews by Kinner *et al* and Zlodre and Fazel.[6 7 12] A scoping review between 2010 and 2020 found 43 publications relating to opioid overdose prevention in justice-involved populations (including correctional and community settings), of which 16 were cohort studies with data linkage.[13] The scoping review identified five core domains regarding opioid overdose prevention: acceptability, accessibility, effectiveness, feasibility and participant overdose risk.[13] The review found evidence of an immediate risk of overdose following release from prison; with the risk factors for opioid overdose including access to drugs, contact with drug using social groups and lack of social and socioeconomic support.[13] Often, this at-risk group had contact with community-based service providers (after their release and before overdose) thereby pointing to opportunities for overdose prevention interventions during this period.[13]

A recent systematic review found that interventions for prisoners with a mental health condition improved the transition process from prison to the community in terms of sustained contact with mental health services.[14] The interventions included in the review differed, for example, in stages of prison release and content, and further evidence is required.[14] Further research of at-risk groups on release from prison may help address their unmet needs, lead to more effective care, reduce negative outcomes and improve public health. This scoping review will focus specifically on studies that use record linkage methodology to investigate drug-related deaths in former prisoners as part of the work programme that is being undertaken by the Administrative Data Research Centre, Northern Ireland.

## Purpose of review

The purpose of this scoping review is to perform an international review of the literature in record linkage studies of drug-related deaths among former adult prisoners. This review will add to current understanding by updating and mapping research evidence in this area, identify and profile former prisoners who are most at risk, and inform potential interventions and future research and policy.

## METHODS AND ANALYSIS

This is a protocol for a scoping review. The framework for conducting scoping reviews by Arksey and O'Malley, in addition to the methodology and guidance developed by the Joanna Briggs Institute (JBI) and the JBI Collaboration will be followed to ensure rigour and transparency in this review.[15 16] Our methodology will follow the first five stages of the framework as detailed below. As recommended by the JBI, population, concept and context (PCC) will be incorporated into the scoping review title and research questions and the inclusion criteria will provide clear details of the PCC.[16] The Preferred Reporting Items for Systematic reviews and Meta-Analyses extension for Scoping Reviews checklist and guidance will be used to report this review.[17]

### Stage 1: identifying the research question

The research questions are:
1. What is the scope of the literature on record linkage studies of drug-related deaths among former adult prisoners who have been released to the community?

2. How is research conducted on this topic?
3. What methodologies are used?
4. What are the findings in relation to mortality?
5. Where are the knowledge gaps on this topic?

## Stage 2: identifying relevant studies

MEDLINE, EMBASE, PsychINFO and Web of Science will be searched for studies from January 2011 to September 2021 using keywords and index headings (modified for searches in each database as required). The start date of 2011 was chosen for this scoping review to summarise the most recent evidence in this field of research. The search terms relate to one of the following three key areas: 'mortality', 'drugs' and 'ex-prisoner' and their variants. The search terms for the strategy were identified from previously published key literature and through discussion between JAC and MD (with assistance from the Subject Librarian for the School of Medicine, Dentistry and Biomedical Sciences in Queen's University Belfast). A search strategy has been developed for MEDLINE by JAC and MD with assistance from the Subject Librarian. This search strategy has been provided with this protocol as supplementary material (see online supplemental appendix 1). JAC and MD will develop further search strategies for other databases to be searched. Final search strategies will be provided with the published review. All reference lists of included studies will be screened for the addition of any further publications. The search strategy will be limited to publications available in English due to resources for translation.

## Stage 3: study selection

JAC will perform all searches in the databases. The results across all of the databases will be combined in Mendeley Reference Manager and any duplicate publications will be removed. Two authors will independently screen all titles and abstracts for eligibility using the inclusion criteria. Any publications not eligible for inclusion will be excluded. If an abstract is not available or there is uncertainty over inclusion, the full publication will be screened. Full publication screening for inclusion will be performed independently by two authors. Any discrepancies between the two authors will be discussed with a third author. Authors of publications will be contacted for more information, if relevant.

Eligibility will be based on the pre-specified inclusion criteria in this protocol. The inclusion and exclusion criteria are as follows:

### Population

The population will include adults (defined as 18 years and older) who have been imprisoned and released to the community. Individuals released from custodial placements such as young offender institutions will be excluded. Individuals remaining in prison custody (eg, prisoners on remand and sentenced prisoners) will be excluded. There will be no exclusion on gender.

### Concept

The key concepts revolve around record linkage of drug-related deaths in adults who have been imprisoned. Included studies must use data linkage (or similar meaning terms) to determine mortality outcomes following release from prison. Studies with no data linkage will be excluded. Only studies reporting cause-specific mortality (ie, drug-related deaths) for either the entire study population or a subset of the study population will be included.

### Context

All geographical locations will be included. The review will include research from peer-review journals. Qualitative studies, commentaries, editorials and conference abstracts will be excluded.

## Stage 4: charting the data

Charting the data will allow extraction of the data from the eligible studies and a draft charting form has been developed to record key information (see online supplemental appendix 2). The development of the form has been informed by relevant previous literature, reviews and the content of RECORD (REporting of studies Conducted using Observational Routinely-collected health Data) statement.[18] This form has been piloted by JAC and IO as part of the protocol development stage. The draft charting form may be refined by authors during the review, as required. The charting form will be tested by at least two authors to ensure that all relevant key information is extracted consistently and accurately. Two authors will independently extract information from the eligible publications using this form. Any discrepancies between the two authors will be discussed with a third author. Authors of publications will be contacted for more information, if necessary.

## Stage 5: collating, summarising and reporting the results

The data will be presented in relation to the purpose of the review (an international review of record linkage studies of drug-related deaths among former adult prisoners) and in the format that will answer the scoping review questions. We will analyse and summarise reports of included studies that used data linkage to determine mortality outcomes following release from prison in terms of, for example, crude mortality rates and standardised mortality ratios. The results will be presented as a map of the data (guided by the PCC inclusion criteria). The extracted data will be summarised and presented in tables. The results will also be organised according to the main conceptual categories (including methodology, key findings and gaps in the research).

## Patient and public involvement

RK is a core member of this review team. RK is the Clinical Director of Healthcare in Prison in Northern Ireland. No additional patient and public involvement.

## DISCUSSION, ETHICS AND DISSEMINATION

This review will identify, map and summarise publicly available sources of research and therefore does not require ethical approval. This work will review the evidence in record linkage studies of drug-related deaths among former adult prisoners and may be used to identify those most at risk, and inform future research and policy. Where possible, time period examined after prison release and information on specific drugs will be reported in relation to drug-related mortality. There are some limitations in this planned scoping review. The search strategy is limited to publications available in English due to resources for translation and the review does not include a search of the grey literature which may limit the interpretation of the findings. The results of the review will inform our empirical investigation of ex-prisoner mortality as well as being shared with key stakeholders, disseminated in a peer-reviewed journal and presented at relevant conferences.

**Acknowledgements** The authors would like to acknowledge Richard Fallis, Subject Librarian for the School of Medicine, Dentistry and Biomedical Sciences in Queen's University Belfast, for assisting in the development of the search terms and search strategy to be used in this scoping review.

**Contributors** JAC, DO'R, RK and MD conceived the scoping review idea. JAC and MD developed the scoping review protocol, scoping review title, research questions and methods. JAC drafted the manuscript. MD edited the first draft of the manuscript. JAC and IO piloted the draft charting form as part of this protocol development. All co-authors reviewed the manuscript and have given final approval for publication.

**Funding** This work is funded by a grant awarded to the ADRC NI by the Economic and Social Research Council (ESRC) (grant number ES/S00744X/1).

**Competing interests** None declared.

**Patient and public involvement** Patients and/or the public were not involved in the design, or conduct, or reporting, or dissemination plans of this research.

**Patient consent for publication** Not applicable.

**Provenance and peer review** Not commissioned; externally peer reviewed.

**ORCID iD**
Janine A Cooper http://orcid.org/0000-0002-8223-3013

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
