## [Reviewer comments · BMJ Open]

ARTICLE DETAILS

TITLE (PROVISIONAL)	Record linkage studies of drug-related deaths among former adult prisoners who have been released to the community: a scoping review protocol
AUTHORS	Cooper, Janine; Onyeka, Ifeoma; O'Reilly, Dermot; Kirk, Richard; Donnelly, Michael

VERSION 1 – REVIEW

REVIEWER	Murphy, Philip N Edge Hill University
REVIEW RETURNED	18-Sep-2021

GENERAL COMMENTS	This is a research topic with the capacity to save former prisoners from premature death. Consequently, it is a very worthwhile project. In my opinion, the literature in this field needs more precise reporting of the specific drugs implicated in premature deaths, where these occur in former prisoners, rather than the use of broad terms such as 'drug related'. Hopefully, the current research team will seek to report this level of detail where possible. Time elapsing before death since last incarceration remains a key variable in this area, and is well represented in the current protocol. I would like to suggest that, where possible, this project record the presence or not of any opiate antagonist intervention following prison discharge. This would provide important evidence for ongoing debates around the availability of 'take-away' naloxone as a preventive measure, and also the licensing of naltrexone by injection for the prevention of opiate use relapse. In my experience, it is unlikely that psychological or behavioural predictors of premature deaths, such as mood measures or measures of hedonistic intent following incarceration, will be found by the researchers, but if such findings are available, their onward dissemination would make an important contribution to this research topic.
---

REVIEWER	Harris, Mark University of New South Wales, School of Public Health and Community Medicin
REVIEW RETURNED	23-Dec-2021

GENERAL COMMENTS	This scoping review protocol of record linkage studies of drug related deaths in prisoners released to the community is appropriately structured. This includes the introduction which describes the previous relevant reviews conducted in the past two
--

	decades. There are, however, some issues that the authors should address:-  1.The review does not include a search of the grey literature. This will significantly limit the interpretation of the findings especially as previous reviews have included grey literature studies. 2.The exclusion of prisoners on remand is a major limitation as this group has been shown to be at risk of drug related death in previous research. 3.An important characteristic of prisoner population at risk of drug related death is ethnicity especially indigenous populations and thus it will be important to capture this in the data extraction template 4.Although this is a scoping review, the review will extract outcomes and these need to be defined in the protocol. 5.The discussion should describe to the limitations of the review.
--	--

VERSION 1 – AUTHOR RESPONSE

Reviewer: 1

Dr. Philip N Murphy, Edge Hill University

Comments to the Author:

This is a research topic with the capacity to save former prisoners from premature death.

Consequently, it is a very worthwhile project. In my opinion, the literature in this field needs more precise reporting of the specific drugs implicated in premature deaths, where these occur in former prisoners, rather than the use of broad terms such as 'drug related'. Hopefully, the current research team will seek to report this level of detail where possible. Time elapsing before death since last incarceration remains a key variable in this area, and is well represented in the current protocol.

Response: We thank the reviewer for this comment. We agree that, where possible, information about specific drugs will be reported in relation to drug-related mortality. We have added a sentence to our protocol to state that we aim to report this level of detail, if available. Text added to section Discussion, ethics and dissemination “Where possible, time period examined after prison release and information on specific drugs will be reported in relation to drug-related mortality.”. Please see track changes. In the data charting form, this will be recorded under ‘Main outcomes - Stated outcome events or summary measures’.

I would like to suggest that, where possible, this project record the presence or not of any opiate antagonist intervention following prison discharge. This would provide important evidence for ongoing debates around the availability of 'take-away' naloxone as a preventive measure, and also the licensing of naltrexone by injection for the prevention of opiate use relapse. In my experience, it is unlikely that psychological or behavioural predictors of premature deaths, such as mood measures or measures of hedonistic intent following incarceration, will be found by the researchers, but if such findings are available, their onward dissemination would make an important contribution to this research topic.

I wish the research team good luck with this project.

Response: We thank the reviewer for raising this important point. However, the focus of our review revolves around record linkage studies and it is unlikely that our search strategy will capture intervention studies. We will note your point as a potential limitation of the review and we will discuss interventions following prison release in the discussion section of our scoping review.

Reviewer: 2

Prof. Mark Harris, University of New South Wales

Comments to the Author:

This scoping review protocol of record linkage studies of drug related deaths in prisoners released to the community is appropriately structured. This includes the introduction which describes the previous relevant reviews conducted in the past two decades. There are, however, some issues that the authors should address:-

1.The review does not include a search of the grey literature. This will significantly limit the interpretation of the findings especially as previous reviews have included grey literature studies.

Response: We thank the reviewer for this comment. We have added limitations under the section 'Discussion, ethics and dissemination'. Text added "There are some limitations in this planned scoping review. The search strategy is limited to publications available in English due to resources for translation and the review does not include a search of the grey literature which may limit the interpretation of the findings.". Please see track changes.

2.The exclusion of prisoners on remand is a major limitation as this group has been shown to be at risk of drug related death in previous research.

Response: The focus of our scoping review is on people released from prison but we will note in the discussion of our scoping review that other prisoner groups (such as prisoners on remand) would benefit from separate review attention.

3.An important characteristic of prisoner population at risk of drug related death is ethnicity especially indigenous populations and thus it will be important to capture this in the data extraction template

Response: We thank the reviewer for this comment. We will discuss mortality in relation to age, gender and race/ethnicity, where possible. This will be captured in the data extraction template (Participants and main results sections).

4.Although this is a scoping review, the review will extract outcomes and these need to be defined in the protocol.

Response: We thank the reviewer for this comment. We have added text "We will analyse and summarise reports of included studies that used data linkage to determine mortality outcomes following release from prison in terms of, for example, crude mortality rates and standardised mortality ratios" under Stage 5: collating, summarising and reporting the results. This will be captured in the data extraction template (Main results).

5.The discussion should describe to the limitations of the review.

Response: We thank the reviewer for this comment. We have added limitations under the section 'Discussion, ethics and dissemination'. Text added "There are some limitations in this planned scoping review. The search strategy is limited to publications available in English due to resources for translation and the review does not include a search of the grey literature which may limit the interpretation of the findings.". Please see track changes.